# Exploring the Structure–Performance Relationship of Sulfonated Polysulfone Proton Exchange Membrane by a Combined Computational and Experimental Approach

**DOI:** 10.3390/polym13060959

**Published:** 2021-03-20

**Authors:** Cataldo Simari, Mario Prejanò, Ernestino Lufrano, Emilia Sicilia, Isabella Nicotera

**Affiliations:** Department of Chemistry and Chemical Technologies—CTC, University of Calabria, Via Pietro Bucci, 87036 Rende, Italy; ernestino.lufrano@unical.it (E.L.); emilia.sicilia@unical.it (E.S.); isabella.nicotera@unical.it (I.N.)

**Keywords:** sulfonated polysulfone, PEMFC, MD, DFT, PFG-NMR, water dynamics

## Abstract

Sulfonated Polysulfone (sPSU) is emerging as a concrete alternative to Nafion ionomer for the development of proton exchange electrolytic membranes for low cost, environmentally friendly and high-performance PEM fuel cells. This ionomer has recently gained great consideration since it can effectively combine large availability on the market, excellent film-forming ability and remarkable thermo-mechanical resistance with interesting proton conductive properties. Despite the great potential, however, the morphological architecture of hydrated sPSU is still unknown. In this study, computational and experimental advanced tools are combined to preliminary describe the relationship between the microstructure of highly sulfonated sPSU (DS = 80%) and its physico-chemical, mechanical and electrochemical features. Computer simulations allowed for describing the architecture and to estimate the structural parameters of the sPSU membrane. Molecular dynamics revealed an interconnected lamellar-like structure for hydrated sPSU, with ionic clusters of about 14–18 Å in diameter corresponding to the hydrophilic sulfonic-acid-containing phase. Water dynamics were investigated by ^1^H Pulsed Field Gradient (PFG) NMR spectroscopy in a wide temperature range (20–120 °C) and the self-diffusion coefficients data were analyzed by a “two-sites” model. It allows to estimate the hydration number in excellent agreement with the theoretical simulation (e.g., about 8 mol H_2_O/mol SO_3_^−^ @ 80 °C). The PEM performance was assessed in terms of dimensional, thermo-mechanical and electrochemical properties by swelling tests, DMA and EIS, respectively. The peculiar microstructure of sPSU provides a wider thermo-mechanical stability in comparison to Nafion, but lower dimensional and conductive features. Nonetheless, the single H_2_/O_2_ fuel cell assembled with sPSU exhibited better features than any earlier published hydrocarbon ionomers, thus opening interesting perspectives toward the design and preparation of high-performing sPSU-based PEMs.

## 1. Introduction

Hydrogen energy and fuel cell technologies play an important role in both environmental and economic issues. Fuel cells can help to reduce the emission of greenhouse gases through efficient and innovative materials and designs. Among them, polymer electrolyte membrane fuel cells (PEMFCs) have been the most promising power source for the electric vehicles due to zero-CO_2_ emission and high efficiency power generation. However, still, there are significant problems prohibiting the large-scale commercialization of this fuel cell technology mainly related to the limited lifetime under practical operational constraints and the high-cost [1].

Research in this field is mainly focused on the design and development of new materials in order to improve the performance and reduce the costs of both electro-catalysts and proton exchange membranes (PEMs), which constitute the MEA (membrane-electrodes-assembly), the heart of a PEMFC.

Regarding the PEMs, the state of the art is still based on Nafion^®^, the perflurosulfonic acid (PFSA) membrane produced by DuPont^TM^ via the copolymerization of an unsaturated perfluoroalkyl sulfonyl fluoride (PSF) with tetrafluoroethylene (TFE). Its price alone is expected to be approximately 30% of the vehicle’s selling price [2], while if we analyze its environmental impact in terms of the so-called global warming potential (GWP), Nafion^®^ material has the largest impact respect to all the other components of a PEMFC stack [3]. This effect can be easily explained if the manufacturing process of the main substances needed for the polymer synthesis is taken into account. Indeed, as they can reside in the atmosphere for thousands of years, GWP values are four order of magnitude higher than that of CO_2_.

Finally, from the point of view of the fuel cell performance, one of the main limitations is that the Nafion membrane cannot be used ‘as is’ over a wide range of temperature and relative humidity (RH) conditions, as those required for automotive applications. Indeed, the strong dependence of its proton conductivity on the hydration level, limits the cell operating temperature between 60 and 80 °C, thus requiring a complex thermal and water management system. On the other hand, one of the main US DOE (Department of Energy) targets for PEMFCs is the operating temperature of the cell, which must be about 120 °C for improving reaction rate at both electrodes, facilitating thermal and water management and enhancing carbon monoxide tolerance of the platinum catalyst at the anode [4].

In light of the above, it is urgent to direct research towards new chemically and mechanically stable, eco-friendly and low-cost proton-conducting polymers. Non-fluorinated polymers, and in particular aromatic hydrocarbon polymers, are proving to be the most promising strategy to conjugate environmental, cost and efficiency requirements for a large-scale deployment of PEMFC technology.

These polymeric materials, as with Nafion, can successfully act as matrices to produce high performance nanocomposites [5]. Some of the most relevant examples are represented by the polyetheretherketone (PEEK) [6,7], poly ether sulfone [8] or the poly(vinyl alcohol) (PVA) [9].

In this panorama, a polymer that deserves some attention and scientific investigation is Polysulfone (PSU). The literature on this polymer for PEMFC applications is rare [10,11,12], and in two our recent works, we proved that it is a valid and promising candidate when used as polymer matrix in the production of nanocomposite electrolyte membranes for both direct hydrogen (DHFCs) and direct methanol fuel cells (DMFCs) [13,14].

The interest in PSU is dictated by considerations about its large availability on the market, the low environmental impact, excellent thermo-mechanical resistance properties, ability to form polymeric films and, of course, its low cost. Clearly, like most of the pristine macromolecules mentioned above, PSU, being hydrophobic and poorly proton-conducting, must be properly functionalized through a sulfonation reaction to be used in the development of a PEM. The synthetic strategies and the selected sulfonating agents can give sulfonated derivatives of Polysulfone (sPSU) with different structural and transport properties. Therefore, a careful analysis of the adopted synthetic procedure cannot be neglected [11,15]. In general, aggressive sulfonation procedures of non-fluorinated polymers are able to achieve derivatives with high ion exchange capacity (IEC), which leads to a significant increase in the proton conductivity of polymer membranes. However, these treatments are often accompanied by excessive swelling, a significant weakening of mechanical strength and thus a product practically unsuitable in a real cell. Finding the optimal compromise between preserving the chemical and mechanical stability of the polymer material and achieving adequate ion transport properties is the key factor [16,17]. The trimethylsilyl chlorosulfonate was chosen as sulfonating agent in our synthetic procedure for its mild sulfonation activity [18], carrying out the reaction at low temperature (about 50 °C) in order to maintain intact the backbone structure. However, the evaluation of new polymers to replace Nafion in PEMFCs can only be addressed through a basic research for a systematic understanding at a fundamental level of the polymer structure, morphological architecture and organization of the water molecules inside the electrolyte membranes. This knowledge, correlated to their physico-chemical, mechanical and conductivity properties, can then be exploited for the preparation of high performance MEA devices. In spite of this, to date and to the best of our knowledge, the microstructure of sPSU has remained still unexplored.

Accordingly, a multi-level computational protocol utilizing DFT calculations combined with molecular-dynamic simulations was employed in the present work to investigate the effect of the temperature on the functional groups structure and stability, as well as their hydration level. Pulse Field Gradient (PFG) NMR spectroscopy has been widely used to get the direct measurements of water self-diffusion coefficients inside the sPSU membrane. The diffusivity data were interpreted in terms of “two sites” model to clarify the water distribution inside the sPSU systems as a function of temperature and water content, which is, to our best knowledge, inedited for this type of system. To definitively assess the practical application of sPSU in PEMFC devices, a deep performance comparison with commercial Nafion was also carried out in terms of themo-mechanical, dimensional and electrochemical properties. The mechanical properties and dimensional stability of the sPSU membrane were evaluated by dynamic mechanical analysis (DMA) and swelling tests, respectively. Finally, the electrochemical features were investigated by electrical impedance spectroscopy (EIS) and single H_2_/O_2_ fuel cell tests, confirming the great potential of sPSU as low-cost and high performing PEMs.

## 2. Materials and Methods

### 2.1. Synthesis Sulfonated Polysulfone and Membrane Preparation

Commercial polysulfone (Lasulf from Lati SPA) was sulfonated according to the procedure reported elsewhere [14]. Briefly, 2 g of PSU was first dried under vacuum at 80 °C for 24 h and then dissolved in anhydrous chloroform under vigorous magnetic stirring at room temperature until a homogeneous solution was obtained. Therefore, trimethylsilyl chlorosulfonate (Aldrich, St. Louis, MO, USA) was added as sulfonating agent (molar ratio sulfonating agent/repetitive units equal to 2.5) and the reaction left for 7 h at 50 °C under reflux to produce a silyl sulfonate polysulfone. Sodium methoxide (Aldrich) was then used to cleave the silyl sulfonate moieties over 1 h to yield the sulfonated polysulfone (sPSU). After precipitation in a bath of ethanol, the sPSU in fine powder was recovered by filtration, vigorously washed with ethanol, and rinsed several times with distilled water prior to be heated at 60 °C in an oven until dry. Robust transparent membrane of sulfonated polysulfone was prepared via solution casting method. Two hundred grams of polymer were completely dissolved in 10 mL of *N*,*N*-dimethylacetamide (DMAc, Aldrich) at room temperature. The resulting homogeneous solution was cast on a petri dish and dried in an oven at 60 °C until complete evaporation of the solvent (dry thickness = 80 μm). The morphological investigation, carried out by Scanning electron microscopy (SEM), revealed the sPSU membrane had dense, compact and homogeneous structure. Also, the absence of any cracks and/or defects indicates good film quality (see Appendix A). Finally, the membrane was converted into the acid form with 1 M H_2_SO_4_ solution (7 h at 50–60 °C), followed by washing several times with boiling deionized water to remove any residual acid. For the as-prepared sulfonated polysulfone, the ion exchange capacity (IEC), determined via titration [7], was 1.56 meq g^−1^, which corresponds to a sulfonation degree (DS) of 80%.

### 2.2. Characterizations

#### 2.2.1. Water Uptake and Thickness Swelling

Water uptake and thickness swelling were determined by monitoring the weight and thickness change of the sPSU membrane under dry and wet states [19]. After drying the membrane in an oven at 60 °C for 24 h, the weight (*m_dry_*) and thickness (*Th_dry_*) were measured. Then, the samples were immersed in distilled water for 24 h at 20 °C and for 2 h at higher temperatures (from 40 to 100 °C each 20 °C). Thereafter, the membranes were picked out, rapidly dried with a paper tissue to eliminate surface water droplets and their mass (*m_wet_*) and thickness (*Th_wet_*) quickly measured. The procedure has been repeated five times for each sample, with an error of circa 2%. Accordingly, the water uptake (*wu*, %) was calculated as:(1)$$wu\;\left(\%\right)=\;\frac{m_{wet}-m_{dry}}{m_{dry}}\times 100$$
while the thickness swelling (*S_T_**_h_*, %) was determined by the following equation:(2)$$S_{Th}\;\left(\%\right)=\;\frac{Th_{wet}-Th_{dry}}{\;Th_{dry}}\times 100$$

#### 2.2.2. NMR (PFG and Relaxometry) Spectroscopy

The ^1^H-NMR measurements were performed on a Bruker AVANCE 300 wide bore spectrometer working at 300 MHz on ^1^H and equipped with a Diff30 Z-diffusion 30 G/cm/A multinuclear probe with substitutable RF inserts. Pulsed field gradient stimulated-echo (PFG-STE) technique [20] was used to directly measure the self-diffusion coefficient (D). This particular sequence allows to measure D in materials characterized by transverse relaxation time (T_2_) considerably shorter than the longitudinal relaxation time (T_1_) and foresees three 90° RF pulses (*π*/2*τ*_1_-*π*/2-*τ_m_*-*π*/2) with two gradient pulses applied after the first and the third RF pulses. At time *τ =* 2*τ*_1_ + *τ_m_* the echo is found. The FT echo decays were analyzed by means of the relevant Stejskal–Tanner expression (Equation (3)):(3)$$I=I_{0}e^{-\beta D}$$
with I and I_0_ representing the intensity/area of a selected resonance peak with and without gradients, respectively, D the self-diffusion coefficient and β the field gradient parameter. This latter is defined by Equation (4):(4)$$\beta =\left[{\left(\gamma g\delta \right)}^{2}\left(\Delta -\;\frac{\delta }{3}\right)\right]$$
where g, δ and Δ are the amplitude, duration and time delay of the gradient field, respectively. For the measurements δ and Δ were kept at 0.8 and 8, respectively, while g ranged between 100–900 G cm^−1^. For the self-diffusion measurements an uncertainty of ~3% was calculated. The detailed experimental procedure to prepare the NMR sample is described elsewhere [21]. During this study, the samples were equilibrated at four different water contents, that are, saturation (38 wt%, maximum membrane swelling), 30 wt%, 20 wt% and 10 wt%, respectively. The measurements were carried out in the range 20–130 °C, with steps of 20 °C and 15 min of equilibration time for each temperature.

#### 2.2.3. Electrochemical Impedance Spectroscopy (EIS) and Fuel Cell Tests

The in-plane proton conductivity was determined by Electrochemical Impedance Spectroscopy (EIS) using a commercial four-electrode cell (BT-112, Scribner Associates Inc.) fitted between the anode and the cathode flow field of a fuel cell test hardware (850C, Scribner Associates Inc., Southern Pines, NC, USA). Temperature and relative humidity (RH) were controlled by the use of a humidification system (Fuel Cells Technologies, Inc. Albuquerque, NM, USA) directly connected to the cell. Here the measurements were performed under different temperature (30, 60, 90 and 120 °C) and relative humidity (30, 50 70 and 90) to simulate the wide operating condition that PEMFCs should withstand. A PGSTAT 30 potentiostat/galvanostat (Methrom Autolab) equipped with an FRA module was used to measure the AC impedance response of the cell. This latter was recorded at OCV in the frequency range between 1 Hz–1 MHz under an oscillating potential of about 10 mV. The impedance spectra were analyzed by NOVA software to extrapolate the electrolyte resistance (*R_el_*) as the high-frequency intercept on the real axis of the Nyquist plot. From its ohmic resistance, the proton conductivity (*σ*) of the electrolytes, expressed in S cm^−1^, were calculated according to Equation (5):(5)$$\sigma =\;\frac{d}{\;R_{el}*A}$$
where *d* is the distance between the electrodes and *A* is the active surface area.

The H_2_/O_2_ fuel cell test were carried out according to the procedure reported elsewhere [22]. Briefly, the MEAs was tested in a 5 cm^2^ single cell in the temperature range 80–110 °C, feeding 200 mL min^−1^ of hydrogen and oxygen to the anode and cathode, respectively, at atmospheric pressure and varying the RH at the anode from 25 until 100%. Reactants stoichiometry was maintained constant during the experiments. Galvanostatic measurements were carried out under steady-state conditions by connecting the cell to a home-made test bench equipped with an electronic load (Fideris, 125 W, 20 V, 5 A). Mass flow controllers (Brooks Instruments, Hatfield, PA, USA) were used to control the feed flow rate on either side of the cell, while the temperature and RH of the cell were measured by appropriate sensors.

### 2.3. Computational Methods

#### 2.3.1. Molecular Models

With the aim to investigate structural features of a sPSU-based membrane, a coupled molecular dynamics (MD) and quantum mechanical (QM) density function theory (DFT) study was carried out. A model of the polymer has been obtained taking into account 270 different monomer units (the monomer unit is shown in Figure 1), in order to both save computational time and to ensure reliability of the model. According to experimental evidence, which indicates a membrane sulfonation level equal to 80%, the model contains 216 -SO_3_ groups.

A well assessed protocol, [23,24,25] based on the General Amber Force Field (GAFF) [26] and RESP method [27], was adopted to extrapolate Lennard-Jones and charges parameters, on previously HF/6-31G* optimized structures of each component of the polymer. The corresponding parameters are provided in AMBER format in the Supplementary Information.

Starting from the equilibrated structures obtained from a preliminary MD study on a single polymer model containing 72 monomers (see Appendix A for further details). The final geometry was obtained adopting the Packmol software [28]. The model was finally filled with TIP3P water molecules and Na^+^ counterions have been added to neutralize the charge, to a total number of 122,873 atoms. For this model, the periodic boundary conditions have been fixed and the final box was of 138 × 110 × 100 Å^3^. A detailed description on the setup of the model is provided in the Appendix A.

#### 2.3.2. Molecular Dynamics Simulations

After a preliminary steepest descent minimization, each system was gradually heated to the selected temperature of 80, 100 and 120 °C, for 25 ns. Additionally, 25 ns of MD simulations were carried out in NVT conditions using the Langevin thermostat. The production phase was performed for 500 ns of MDs under the following conditions: Integration step of 2 fs coupling SHAKE algorithm [29] and NPT ensemble at 1 bar pressure using the Berendsen barostat with a time constant τp = 2.0 ps. The Particle Mesh Ewald summation method [30] was employed for the electrostatic potential and the long-range electrostatic interactions were calculated with 12 Å cut-off distance. All the simulations were performed using the AMBER16 package [31]. The entire trajectories were clustered and analyzed via cpptraj software [32].

#### 2.3.3. DFT Calculations

A model system containing two neighbor monomers was selected from MD simulations and was fully optimized at DFT level of theory, adopting the B3P86 [33] functional and the 6-31+G(d) basis set for H, C, O and S atoms. The nature of minimum of the optimized structure was confirmed by vibrational frequencies (all positive) calculation. All the calculations were carried out adopting the Gaussian09 D.01 software [34]. Non-Covalent interactions were plotted via NCIPLOT suite [35], in order to identify regions in which the favorable interaction occurs. The final model presented 150 atoms and a total charge equal to −2.

## 3. Results & Discussion

### 3.1. Computational Studies

The results of the computational investigation, carried out using MD simulations, of hydrated sPSU-based membrane will be discussed with a particular focus on the adopted biggest model. Due to the lack of information about the internal rearrangement characterizing the structure of the sPSU-membrane, we decided to carry out all-atoms classical MD simulations in order to investigate conformational behavior of the membrane, with the aim to deeply understand how this can affect the efficiency in PEM-FCs. It is worth noting, indeed, that the theoretical modelling of polymers, nowadays, plays a crucial role in the understanding of chemico-physical properties of polymers. The information obtained from simulations can be further adopted in the design of new materials and the discovery of new synthetic routes. To this purpose, it has been demonstrated as many levels of theory can provide useful insights in polymers science and in the further development of FCs, in particular, adopting modern computational chemistry tools, like Coarse-Grained (CG) simulations and fractal theory (see refs [36,37,38] as recent examples). The MDs simulations were carried out at different temperatures, aiming at investigating several possible conformational scenarios promoted by different thermal conditions. As, usually, a PEMFC can work at a temperature higher than 80 °C, for this reason, three temperatures, 80 °C, 100 °C and 120 °C, were selected. Furthermore, the range between 80 °C and 120 °C, represents the interval in which the most relevant insights were achieved by experimental techniques and discussed in next paragraphs. The structures sampled during the calculation of the trajectories, at 80 °C, 100 °C and 120 °C are reported in Figure 2. These structures are representative of more than 50% of the calculated conformations, obtained for each frame of the MD simulation after geometrical hierarchical clustering of the trajectories [39] (see Appendix A), indicating that the polymer presented conformational homogeneity for more than half of the simulation with respect to the initial structure depicted in Appendix A. In particular, backbone–backbone interactions between the aromatic rings of the polymer and closer distances between intra- and inter-molecular SO_3_ groups during the MD simulation were observed. As a direct consequence of this molecular organization, the formation of spread and shared clusters of water molecules between the sulfonate moieties in formed channels and pores occurs. Generally, indeed, in all the sampled structures, the sulfonate groups (colored in yellow and red in Figure 2) are located at the interface between the water molecules (in grey in Figure 2) and the hydrophobic region (colored in blue, cyan and green in Figure 2) characterized by the polymer’s aromatic rings, as similarly observed in MDs for Nafion^®^-based membranes [40,41,42]. Detailed results, like Root Mean Square Deviation trend, of the Molecular Dynamics simulations are reported in Appendix A. Pair Correlation Functions (PCFs) were calculated along the trajectories in order to obtain further information about the polymer configuration and the hydration of its functional groups.

The main results are shown in Figure 3. In particular, for what concerns the intermolecular and intramolecular sulfur-sulfur PCF of SO_3_ groups (Figure 3A), the results show several peaks: The first one centered at about 6.08 Å (80 °C) with decreasing (5.81 Å) and increasing shifts (6.52 Å) at 100 and 120 °C, respectively, and the second broader peak at a distance of 8.20 Å, for all the three different temperatures. The two maxima are those belonging to intramolecular sulfur-sulfur pair. In all the three cases, a broader peak appeared over 12.0 Å and in proximity of 18.0 Å. For the temperature of 80 °C, the magnitude of the peaks resulted higher with respect to the others, and in particular for the maximum at 18.0 Å. This effect can be attributed to the fact that the temperature increase, promoting the higher mobility of the water molecules in the hydrophilic channels, causes a variation of the pores size in which the water molecules are located. The presence of regions within the range 5.50–19.20 Å highlights the formation of hydrogen bond networks of the SO_3_ groups with the water molecules, generating an internal morphology of channels and pores, fundamental for a PEM’s efficiency. The average distance between the SO_3_ groups and their correlated pore size was measured to be equal to 14 ± 2 Å, 15 ± 2 Å and 18 ± 3 Å, for systems at 120 °C, 100 °C and 80 °C respectively, confirming the efficient molecular architecture of sPSU-based membrane as previously mentioned. Additional support comes from the analysis of the PCF calculated for the pair of oxygens of the SO_3_ group and the oxygen of water molecules (Figure 3B). For each adopted temperature, a first intense maximum was localized at a length of 2.75 Å, ascribable to the first hydration shell of the functional group that is in excellent agreement with the acceptor-donor distance requested for the hydrogen bond formation. A second lower peak, characterizing the second hydration shell, was registered at a distance of 4.84 Å, similar to Nafion-based membranes [40,41,43]. A weak but constant decrease of the magnitude of the peaks was registered as function of the temperature increasing and then, kinetic energy that can be related to the enhanced translational movements of the water molecules in proximity of the sulfonate groups. This last trend was further observed analyzing the variation of the hydration number of the SO_3_ group as a function of the hydration sphere, reported in Figure 3C. The average number of water molecules for each SO_3_ group was considered as a function of the increasing hydration sphere radius, with 0.0 < r(Å) < 5.0. The plot highlights as the temperature affects the number of water molecules around the SO_3_ groups, coherently with what discussed for PCF and shown in Figure 3A,B. In fact, a decreasing hydration number was detected, in proximity of 3.50 Å and we calculated that 8.1 (80 °C), 6.5 (100 °C) and 5.1 (120 °C) H_2_O molecules are present for each SO_3_ group.

Additional information about the internal configuration of the hydrophobic portion of the monomer were obtained calculating the PCFs among the selected carbon atoms of the aromatic rings composing the monomer (see Appendix A). Each of the four rings shows the tendency to establish π-π interactions with other rings, despite the variation of the temperature. In particular, the most intense peaks have been registered for the first ring, at 2.87 Å, 3.61 Å (R1–R2) and 5.05 Å (R1–R4), values coherent with the equilibrium distance of π-π stacking, as reported in Appendix A. All the rings present this type of interaction with at least another aromatic ring.

Summarizing, from an inspection and the PCFs discussed above it can be argued that for the sPSU hydrated polymer exists a phase separation, similar to that observed for lamellar-like microstructure. The model that we adopted can be considered as representative of one of the possible lames that can generate a more complexed microstructure. In this condition, the water molecules are engaged in hydrogen bond networks thanks to the presence of well-oriented hydrophilic portion of the material (containing sulfonic groups) while the stability of the internal architecture of the polymer is maintained by the interactions between the backbones of the hydrophobic counterpart (aromatic rings). The pore sizes that were calculated are in agreement with those reported for other membranes, such as poly(ether ketone) and poly(phenylene)-sulfone [44,45]. The results obtained from the analysis of structural and dynamical parameters, such as radial distribution function and hydration number, have been adopted in the experimental investigation that follows in the next paragraph. Additionally, the results presented above can be helpful for the set-up of more accurate force fields in order to carry out both classical and CG-MDs, for the study of sPSU-based membranes via models with a higher number of atoms, usually more accurate for obtaining relevant thermodynamic parameters.

In order to verify the effective stability of the polymer configuration observed during the simulations presented above, a model system selected from the trajectory of the MD simulation performed at 80 °C (see Appendix A) was examined at quantum mechanical DFT level of theory. A focus on the hydrogen bond network between two SO_3_ groups obtained after the QM optimization and the analysis of non-covalent interactions (NCIs) is reported in Figure 4. With this last tool, in particular, it is possible to visualize all NCIs occurring between the atoms, highlighting attractive, van der Waals and repulsive interactions as blue, green and red surfaces, respectively. In the selected cluster, the 10 water molecules establish strong H-bond interactions, as evidenced by positive values of the reduced gradient density (blue surfaces) with each other and with the sulfonate moieties. The green areas, as specified above, can be ascribed to the existence of non-covalent interactions that further stabilize the network of interacting species. Comparing the distances before and after DFT optimization, it is possible to note that the relative hydrogen bond distances decrease below the values of 2.00 Å, as depicted in Figure 4. In particular, some water molecules located in the middle of the cluster act as double donor and double acceptor of H-bond and establish new H-bond interactions with the neighbor atoms that were not registered in MD simulations. In the case of w7, new H-bonds were registered both with oxygens of SO_3_ groups (O3-w7, from 3.50 Å to 1.72 Å, and O6-w7, from 3.20 Å to 1.96 Å) and with other waters (w2-w7, from 3.27 Å to 1.68 Å, and w5-w7, from 3.26 Å to 1.89 Å). After the optimization, furthermore, the S atoms move closer (6.68 Å) with respect to the initial distance observed in the MD (7.89 Å), due to the strong attractive interactions mediated by the water molecules that favor the new displacement of the SO_3_ groups. The illustrated behavior further emphasizes the good membrane predisposition to an effective ion transport, an aspect still under investigation that will be addressed in future publication.

### 3.2. PFG-NMR Investigation and the “Two-Sites” Model

A dee investigation on the water molecular dynamics and distribution within the ionic clusters of sPSU was performed by NMR spectroscopy, and in particular by the ^1^H-Pulse Field Gradient (PFG) method, which allows the direct measurements of the self-diffusion coefficient (D) of water confined in the membrane. Measurements were conducted on membranes swelled at various water contents (water uptakes) and in a wide temperature range (20–130 °C). Figure 5 reports the diffusivity data obtained at four different water contents, i.e., saturation (38 wt%, maximum membrane swelling), 30 wt%, 20 wt% and 10 wt%, respectively, in order to explore the transport features of sPSU in the various humidification states. Clearly, *D* is strongly correlated to the membrane’s water content, proportionally increasing with it [46,47]. On the other hand, the abrupt collapse of *D* observed above 60–80 °C is simply amenable to water evaporation: At high temperatures, the ^1^H-signal originates almost exclusively from the so-called bound-water molecules, namely the ones strongly interacting with hydrophilic ${\mathrm{SO}}_{3}^{-}$ groups of the polymer, since the most of free-water is evaporated from the membrane. Accordingly, at 100 and 120 °C, the self-diffusion coefficients converge to an identical value independently from the initial water uptake, suggesting that each membrane has reached an equal fraction of “bound water”.

These data were processed on the basis of the “two-site” model, which was successfully used to describe the water distribution in Nafion membrane. [48]. The model takes into account the two water fraction, “free” and “bound”, coexisting inside the ionic clusters of PEMs [49,50,51] and that are in rapid exchange compared with the NMR times involved in the experiment [52]. Therefore, we observe only one broad NMR signal, and the measured diffusion coefficient *D*, is an average on the mobility of water molecules in the free (*D_f_*) and bound (*D_b_*) states, weighted on the respective mass fractions, according to the following equation (Equation (6)):(6)$$D=\chi_{f}\;D_{f}+\chi_{b}D_{b}$$

*χ_b_* represents the mass fraction of water molecules directly involved in the hydration of -${\mathrm{SO}}_{3}^{-}$ groups, while *χ_f_* represents the mass fraction of free-water molecules (or known as bulk-like water). Their sum is the total amount of water absorbed by the membrane (water uptake): $\chi_{f}+\chi_{b}=wu$.

Taking into account that *D_b_* is neglectable compared to *D_f_*, and by expressing the free-water diffusion coefficient as function of the bulk-water diffusion per an obstruction factor *φ* (*D_f_ =*
*φ D*_0_) (for further details on the “two site” model see Nicotera et al. [48]), the equation 7 can be simplified in accordance with the Equation (7):(7)$$D\approx \chi_{f}\;\varphi \;D_{0}$$

The obstruction factor is a geometrical parameter that ranges between zero (no diffusion) and 1 (free state), depending on both the freedom degrees of the diffusing species (interactions) and the tortuosity of the diffusion path (network structure) [53,54,55]. As with Nafion, for this polymer we try to estimate the obstruction factor through the Faxen’s equation (Equation (8)), based on the assumption that the water molecules in the polymer are assimilated to spheres diffusing between two rigid solid plates [56,57]:(8)$$\varphi =1-\frac{2R_{H}}{d_{w}}$$
where *R_H_* is the hydrodynamic radius of the particle (~0.1 nm for water), and *d_w_* is the distance between the plates, which should correspond to the diameter of hydrophilic channels in sulfonated polysulfone.

The complex multi-level computational study illustrated above proposes a fibrillar-like structure for the sPSU polymer, with hydrated ionic clusters having a diameter between 14 and 18 Å depending on the water content. The obstruction factor calculated applying the Faxen’s formula varies from 0.89 under fully humidification, to 0.86 at an almost dry state. Such values are lower than that estimated for Nafion (it was estimated to be 0.92), in agreement with narrower ionic clusters and likely more branched in this sulfonated polysulfone.

The free-water mass fraction (*χ_f_*) in the sPSU membrane at 20 °C was calculated by the Equation (8), correlating it directly with the water content, i.e., $\chi_{f}=\frac{D}{\varphi \;D_{0}}\times wu$, using a *φ* value of 0.89 for higher *wu*, and 0.86 for lower *wu*. Clearly the bound-water fraction is simply: *χ_b_* = *wu* − *χ_f_*.

In order to achieve the free- and bound- water mass fractions also at higher temperatures, the water content at each temperature was estimated from the correspondent area of the ^1^H-NMR signal (see Appendix A).

The results, from 20 up to 100 °C, and for all the starting *wu* of the membrane, are plotted as histograms in the Figure 6.

These data give a picture of the water content evolution and distribution inside the hydrophilic clusters of sPSU membrane:By increasing the temperature, the total amount of water content gradually decreases due to the evaporation (as already seen by the diffusivity and ^1^H-NMR signal evolution),The membrane swelling (going from 10 wt% up to 38 wt% of water uptake) produces a continuous redistribution of the water molecules among the hydration spheres of the sulfonic groups and the bulk-water, with a progressive growth of both *χ_f_* and *χ_b_*,Most of the water confined in the membrane is in a “bound state”, i.e., involved in the hydration shell of the polymer’s sulfonic groups since *χ_f_* is much lower than χ,*χ_f_* gradually decreases during heating and practically collapses above 80 °C, and at 100 °C there is only hydration water.

At this point we can estimate the hydration number to the sulfonic groups, $h_{{\mathrm{SO}}_{3}^{-}}$, at each temperature and membrane swelling condition, accordingly to Equation (9):(9)$$h_{{\mathrm{SO}}_{3}^{-}}=\;\frac{\chi_{b}}{\;MW_{H_{2}O}}\;\times \frac{10}{1.56\;\mathrm{mmol}\;{\mathrm{SO}}_{3}^{-}}$$
where $MW_{H_{2}O}$ is the molecular weight of water (18 g/mol), therefore the ratio $\frac{\chi_{b}\times 10}{\;MW_{H_{2}O}}$ represents the mmoles of bound-water, while the ${\mathrm{SO}}_{3}^{-}$ groups present in the membrane are 1.56 mmoles per 1 gr of polymer (see IEC evaluation).

Figure 7 shows the plot of the hydration numbers as function of temperature and initial uptake condition of the membrane. In the graph, the data at 120 °C are also reported. It was calculated by considering the area of the ^1^H-spectra, and assuming that the signal is generating only by the hydration water.

The number of water molecules hydrating the sulfonic groups ranges between 10.2 in the maximum hydration state, and 2.6 in quasi-anhydride condition. To our best knowledge, up to now, no other papers report an estimation of the hydration numbers for the sulfonated polysulfone.

In comparison to Nafion they are slightly lower [48,58,59], likely related to both the remarkably higher IEC of the sPSU (1.56 meq g^−1^ vs. 0.90 meq g^−1^ of Nafion), and the structuring of the polymer network. It is worth noting the gradual reduction of the hydration shell of sulfonic groups up to 80 °C, which suddenly fall at higher temperatures. Under nearly dry conditions, the hydration number became almost 3.3 at 100 °C and 2.6 at 120 °C, no matter the initial uptake. This is coherent with the remarkable decrease in the ionic cluster diameter evidenced by the computational analysis. Additionally, the theoretical simulation had estimated a hydration number at 80 °C of 8.1 (under conditions of maximum water quantity), which is very close to the value of 7.8 obtained through the two-site model applied to experimental diffusion data. This excellent agreement strongly confirms the validity of our approach and gives further strength to the morphological features here proposed for sulfonated polysulfone. We must specify, on the other hand, that instead a comparison with the hydration numbers calculated in MD at 100 and 120 °C (6.5 and 5.1, respectively), is not appropriate because the membrane conditions are de facto very different: While computational analysis is carried out in steady-state conditions, in the PFG-NMR experiments, the membrane at 100 °C undergoes a brusque water evaporation (look at D behavior for the *wu* = 38 wt% of Figure 5), therefore the total amount of water reduces drastically.

### 3.3. Mechanical Properties, Dimensional Stability and Electrochemical Performance of sPSU Membrane

An overview of the main operational features as thermo-mechanical, dimensional and electrochemical properties, of the PSU-based membrane, are provided and compared to a standard commercial membrane, namely Nafion 212. Figure 8 shows the temperature evolution of the storage modulus (*E′*) and the dumping factor (*tan δ*) of sPSU and Nafion 212 membranes, respectively. The hydrocarbon membrane shows an impressive mechanical stability without dimensional changes and deterioration until 180 °C, while the *E′* of Nafion starts decreasing at circa 90 °C. This is further confirmed by the *tan δ* profiles, which show a single temperature transition (130 °C for Nafion and 190 for sPSU), ascribable to the *T_g_* of their ionic clusters. Accordingly, sPSU polymer offers the advantage of a much wider thermo-mechanical stability, being able to withstand higher working temperatures and more severe stress conditions compared to Nafion.

For a proton exchange membrane, the water uptake typically represents a double-edged sword: From one side, water molecules favor a more efficient proton conduction, but to the other side an excessive water content generally leads to a harmful fuel crossover and deterioration of the membrane mechanical properties. Accordingly, the water absorption capacity and dimensional stability of sPSU have to be properly evaluated in view of practical application in fuel cell system. Figure 9 shows the swelling behavior (i.e., water absorption and thickness variation) of the sPSU membrane as function of the temperature. Again, the data are compared with those of the benchmark. As can be clearly seen, up to 60 °C, sPSU is able to maintain good swelling properties albeit showing higher *wu* compared to Nafion. This is likely ascribable to the particular microstructure of the sPSU resulting from the sulfonic groups strictly localized on the rigid aromatic backbone, which allows the absorption of a large amount of water while making the polymer less susceptible to dimensional variation. However, above 80 °C, an excessive water uptake together with a massive dimensional change is observed. This behavior is commonly observed in non-fluorinated polymers, such as sPEEK, sPPEKK and sPAEKK [60,61,62], and represents one of the main drawbacks, which still limits their use as PEMs in high-temperature fuel cells. Nonetheless, we have recently demonstrated that by reducing the sulfonation degree of sPSU (therefore also its IEC), it can be successfully used as polymer matrix to produce nanocomposite PEMs membranes with outstanding dimensional stability (in a wide range of operating conditions) if an appropriate filler is added and homogeneously dispersed, even providing better electrochemical performance than Nafion [13,14].

The temperature-dependence of the proton conductivity of sPSU membrane is showed in the Arrhenius plot of Figure 10, for various RH% conditions. Some representative sigma values are reported in Table 1 and compared to that of Nafion 212. The activation energies (*E_a_*) calculated by the fitting analysis to the experimental data are also reported at each relative humidity condition. Clearly, the conductivity enhances by increasing the temperature and humidification, but the values remains much lower than the benchmark although the higher IEC of the sPSU: It reaches a maximum value of 78.15 mS cm^−1^ at 120 °C and 90% RH, which is about 1.5-fold lower than the benchmark (127.90 mS cm^−1^). This expected outcome suggests as the different structural characteristics of PFSA- and hydrocarbon-based membranes have a great influence on proton conductivity. In fact, the lower hydrophilic/hydrophobic phase separation in sPSU produces narrower nanochannels, poor connectivity between adjacent proton conductive regions and a larger number of dead-end protons paths respect to Nafion. The phenomenon is well described for other non-fluorinated ionomers [63] such as sPEEK, sPES and chitosan membranes for which the conductivity ranges between 60–32 mS cm^−1^ at high temperatures (i.e., 100–120 °C) and 100% RH [46,64,65], therefore somewhat lower than sPSU. The conduction barriers, estimated in the form of activation energies (*E_a_*), are generally higher than Nafion 212 for the aforementioned structural factors, even if they have the same trend respect to the membrane’s humidification condition. At low RH%, the activation energy ranges between 0.33–0.20 eV, which are the typical values for proton transport via Grotthuss mechanism [66]. Increasing the water content, a progressive shift towards lower energy values is observed, and for instance at 90% RH the *E_a_* is 0.14 and 0.12 eV for sPSU and Nafion 212, respectively, describing the low energy-barrier vehicular mechanism for the protons [67]. In a nutshell, although differing in microstructure, both the bulk membranes possess similar intrinsic conductive features.

Finally, single H_2_/O_2_ fuel tests of a MEA based on this sPSU membrane were performed at various temperature and RH conditions. Figure 11 illustrates the performance comparison of single cells assembled with the sPSU and Nafion 212 membranes operating at different temperature and humidification conditions, i.e., 80 °C/30–100% RH and 110 °C/25% RH. Clearly, MEA based on Nafion 212 benchmark membrane was running under the same conditions. The membranes thickness and preparation process of MEAs were kept the same for the two membranes. The open-circuit voltage (OCV) and peak power density values extracted from the polarization and power density curves are summarized in Table 2, where a performance comparison with similar non-fluorinated polymer electrolytes is also provided. The OCV of sPSU is always higher than the benchmark, suggesting a more effective barrier to the fuel’s crossover. In particular, the OCV for the sPSU membrane was roughly 0.90–0.92 V depending on the operating conditions, indicating a very small amount of H_2_ gas permeability from the anode to the cathode through the non-fluorinated membrane. On the other hand, the maximum current density of sPSU is basically more than 50% lower than Nafion 212. For instance, at 80 °C and 100% RH the peak power density is 387.9 and 682.6 mW cm^−2^, respectively. This is clearly amenable to the higher ohmic resistance of sPSU in comparison to Nafion 212. However, compared to earlier published hydrocarbon-based PEMs [65,68,69,70,71], sPSU is much more promising, although it is clear that it shows great limitations when used “bare” and with a high sulfonation degree. It can instead represent a very interesting polymer matrix for developing innovative, low-cost, environmentally friendly nanocomposite membranes, thus opening interesting perspectives toward the preparation of cost effective and low environmental impact PEMs. Finally, it is well known that the fuel cell performance is closely related to the water content. Indeed, more severe operating conditions, i.e., 110 °C and 25% RH, lead to a slight decrease in the maximum power density of both membranes. Likely, the poor ability to retain water of both sPSU and Nafion membranes produces a conspicuous increasing of the ohmic resistance, which reduces the cell performance.

## 4. Conclusions

For the first time, the structure–performance relationship in hydrated sulfonated polysulfone (sPSU) membranes was studied in detail by combining computational techniques and experimental methods. All-atoms molecular dynamics simulations, performed in the temperature range 80–120 °C, revealed a clear micro-phase separation between hydrophilic/hydrophobic domains in sPSU, with sulfonic acid groups placed at the interface between the water molecules and the hydrophobic region, which is instead formed by the aromatic rings of the polymer backbone, with a lamellar-like microstructure. The atomic structures were characterized by various pair correlation functions showing that water-filled channels are formed among the hydrophilic regions of the sPSU, the size of which ranges from 14 Å and 18 Å. Quantum mechanics calculations carried out by means of density functional theory, in addition, emphasized the importance of the hydrogen bond networks established between clusters of water molecules and the SO_3_ groups of the polymer. Direct investigation of water molecular dynamics in sPSU membrane was carried out by PFG-NMR spectroscopy as a function of the temperature and water content. Consequently, diffusivity data were elaborated according to the “two-sites” model for a quantitative estimation of free and bound water fraction in sPSU. Most of the water confined in the narrow ionic cluster is into “bound state”, i.e., directly involved in the hydration shell of the polymer’s sulfonic groups, and hydration number was found to range between 10.2 in the maximum hydration state, and 2.6 in quasi-anhydride condition. Worth note, excellent agreement was found between computational/experimental hydration numbers if similar membrane conditions are taken under consideration. From the performance comparison with Nafion, it clearly emerged that sPSU can effectively tolerate higher working temperatures and more severe stress conditions, demonstrating outstanding thermo-mechanical properties. On the other side, the excessive water uptake and dimensional variation above 80 °C as well as the poor proton conductivity are severe drawbacks that limit its use as PEMs in high-temperature fuel cells. Nonetheless, sPSU yields a maximum proton conductivity of 78 mS cm^−1^ at 120 °C and 90% RH, which is quite high in comparison with earlier published hydrocarbon-based PEMs. Similarly, the H_2_/O_2_ fuel cell performance of sPSU exceeds the one of analogs non-fluorinated polymers, thus opening exciting perspectives toward the preparation of innovative, low-cost and high-performing nanocomposite membranes. In this regard, the present work may drive the future design and development of cost effective and low environmental impact sPSU-based PEMs.

## Figures and Tables

**Figure 1 polymers-13-00959-f001:**
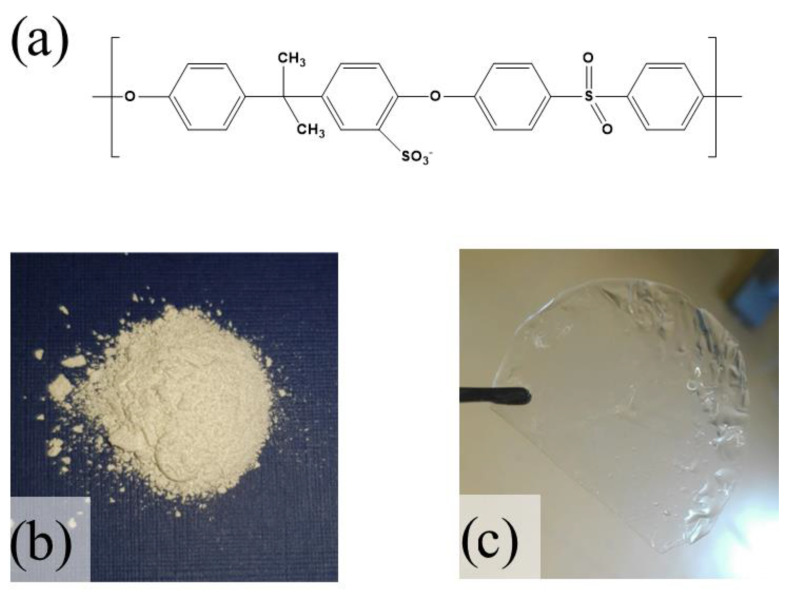
(**a**) Chemical structure of sulfonated polysulfone. Pictures of (**b**) powdered sulfonated polysulfone (sPSU) and (**c**) sPSU membrane.

**Figure 2 polymers-13-00959-f002:**
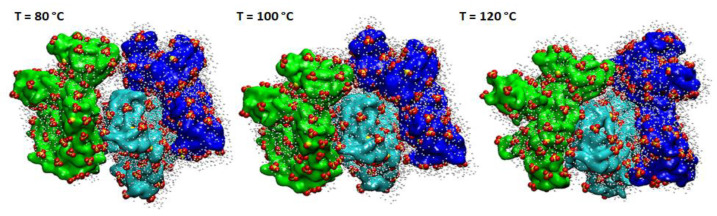
Sampled structures collected from the sPSU-based membrane MD simulations. For clarity, the backbones are represented in green, blue and cyan, the -SO_3_ groups in yellow/red (S and O, respectively) and the water molecules in direct contact with the -SO_3_ in grey.

**Figure 3 polymers-13-00959-f003:**
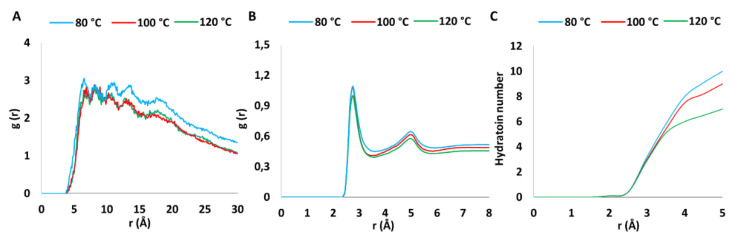
Pair correlation functions between the (**A**) sulfur-sulfur of SO_3_ groups and (**B**) the SO_3_ oxygens—oxygen of water molecules, (**C**) variation of the hydration number around SO_3_ groups as a function of the increasing hydration sphere radius, at different temperatures.

**Figure 4 polymers-13-00959-f004:**
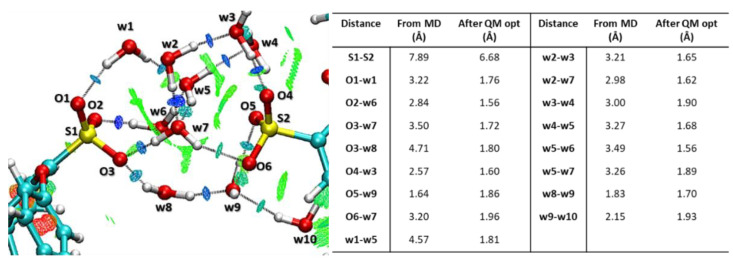
BP86/6-31+G(d,p) optimized geometry of model adopted to study the stability of PSU-polymer. The distances regarding hydrogen bond and the calculated non-bonding interactions (blue and green surfaces) are reported.

**Figure 5 polymers-13-00959-f005:**
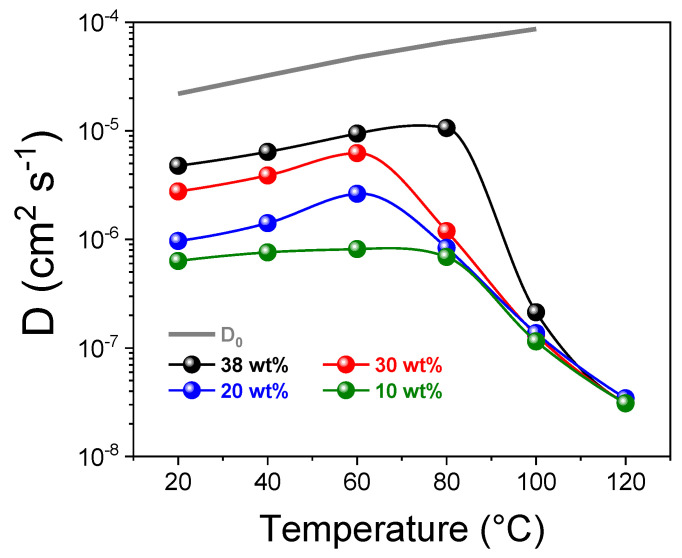
Temperature evolution, from 20 to 120 °C, of the ^1^H self-diffusion coefficients of water sPSU membrane swelled at four water uptakes (saturation, 30%, 20 wt% and 10 wt%). The pure-water diffusion coefficients (D_0_) are also plotted as grey-line.

**Figure 6 polymers-13-00959-f006:**
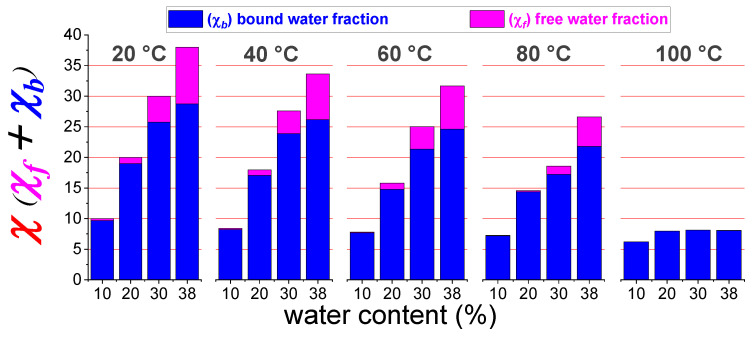
Free- and bound- water mass fractions calculated by the two-model sites equation ($\chi_{f}=\frac{D}{\varphi \;D_{0}}\times wu$; *χ_b_* = *wu* − *χ_f_*) for sPSU membranes at different initial water uptake (*wu*) and in the temperature range 20–100 °C.

**Figure 7 polymers-13-00959-f007:**
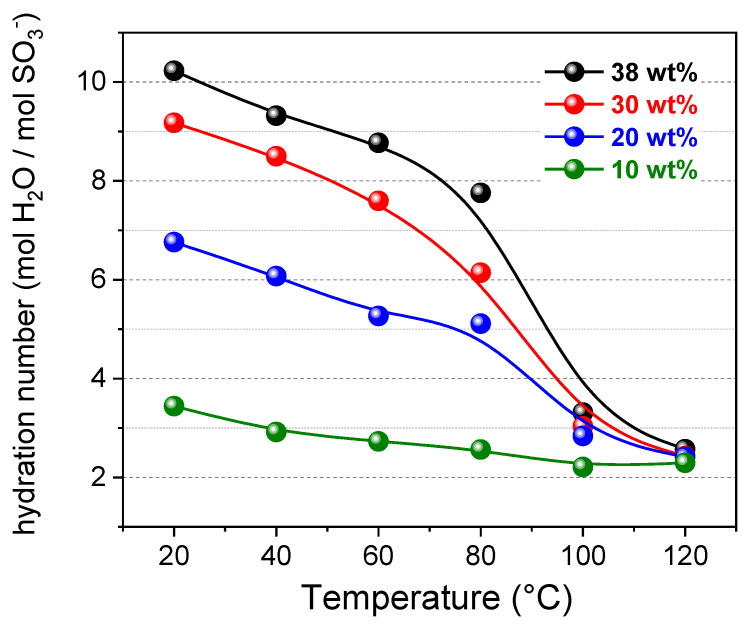
Hydration number for the sPSU membrane at different initial water uptakes (from saturation to 10 wt%) and in the temperature range 20–120 °C.

**Figure 8 polymers-13-00959-f008:**
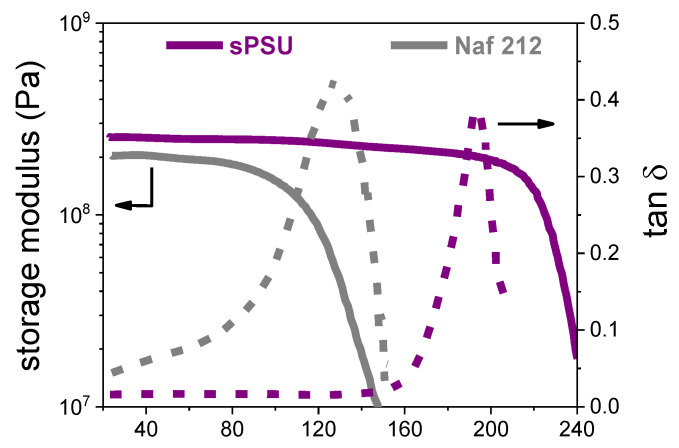
Storage modulus (*E′*) and *tan δ* for sPSU and Nafion 212 membranes as a function of the temperature.

**Figure 9 polymers-13-00959-f009:**
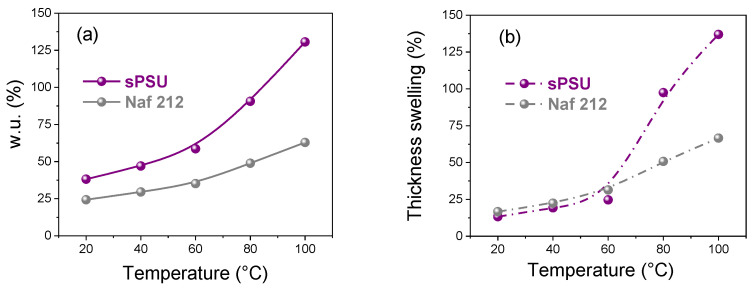
Temperature evolution, from 20 to 100 °C, of the (**a**) water uptake (*wu*) and (**b**) the thickness swelling (*S_Th_*) for sPSU and Nafion 212.

**Figure 10 polymers-13-00959-f010:**
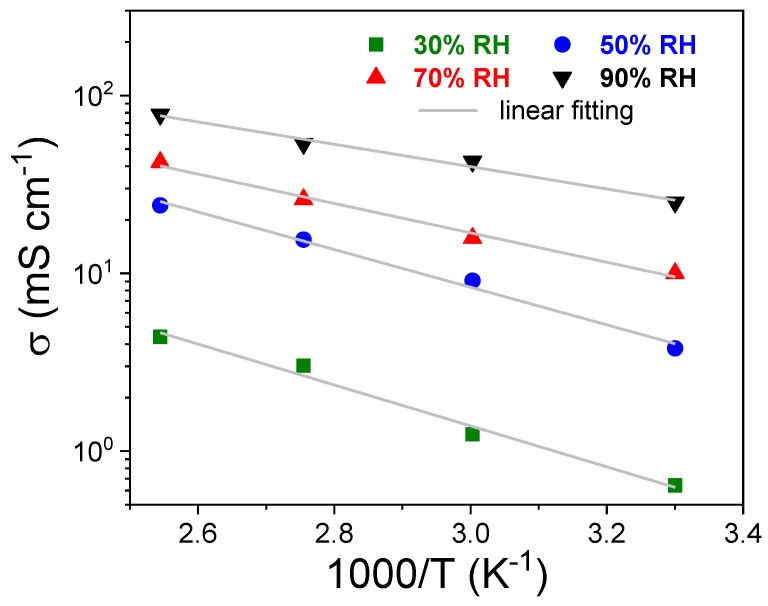
Arrhenius plot for the proton conductivity of sPSU membrane at 30, 50, 70 and 100% RH.

**Figure 11 polymers-13-00959-f011:**
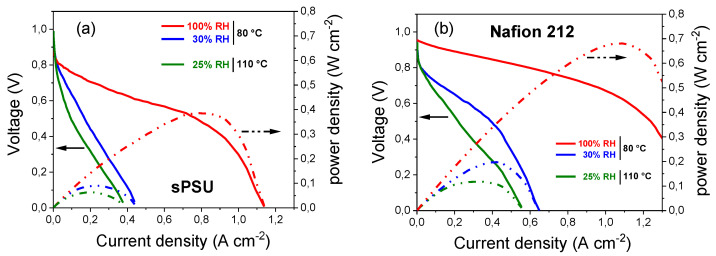
Single H_2_/O_2_ cell performances of (**a**) sPSU and (**b**) Nafion 212 membranes under various temperature and humidity conditions.

**Table 1 polymers-13-00959-t001:** Humidity dependence of proton conductivity and activation energy for sPSU and Nafion membranes.

RH [%]	sPSU			Nafion 212		
	σ @ 90 °C [mS cm^−1^]	σ @ 120 °C [mS cm^−1^]	*Ea* [eV]	σ @ 90 °C [mS cm^−1^]	σ @ 120 °C [mS cm^−1^]	*Ea* [eV]
30	3.0 ± 0.1	4.39 ± 0.1	0.33 ± 0.03	8.8 ± 0. 2	17.4 ± 0.3	0.23 ± 0.02
50	15.5 ± 0.1	24.10 ± 0.2	0.30 ± 0.02	24.4 ± 0.2	38.6 ± 0.4	0.21 ± 0.03
70	26.0 ± 0.2	42.19 ± 0.2	0.21 ± 0.01	48.6 ± 0.3	72.95 ± 0.4	0.17 ± 0.02
90	53.5 ± 0.2	78.2 ± 0.3	0.14 ± 0.02	91.1 ± 0.4	127.9 ± 0.6	0.12 ± 0.01

**Table 2 polymers-13-00959-t002:** State of the art performance for different membranes integrated in H_2_/O_2_ fuel cell.

Membrane	T	RH	OCV	Max Power Density	Ref.
	[°C]	[%]	[V]	[mW cm^−2^]	
**sPSU_80_**	80	100	0.917	388	This work
	80	30	0.908	91	
	110	25	0.899	64	
**Naf 212**	80	100	0.902	683	
	80	30	0.901	197	
	110	25	0.890	118	
**SFMC**	70	100	0.980	182	[68]
**sPEEK**	80	100	0.855	255	[69]
**sPES**	80	100	0.850	66	[70]
**Chitosan**	120	20	0.959	57	[65]
**PBI**	120	dry	0.884	202	[71]

## Data Availability

Data is contained within the article or Appendix A.

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
