# Peer review of "Exploring the Structure–Performance Relationship of Sulfonated Polysulfone Proton Exchange Membrane by a Combined Computational and Experimental Approach"

_polymers, 2021, doi:10.3390/polym13060959_

Round 1

Reviewer 1 Report

Hydrogen as an energy resource can make life significantly more eco-friendly. A hydrogen fuel cell works based on electrochemistry, by passing hydrogen through the anode and oxygen through the cathode of the cell. Proton exchange membrane fuel cell (PEMFC) which includes a membrane, and two electrodes, has grown up with huge attraction because of its simple operation and fuel availability. The proton exchange membrane fuel cells (PEMFCs) are promising energy devices for stationary and mobile applications because of high power density, high efficiency, low operating temperature, low emissions, low noise, and great environmental compatibility. The PEMFCs are composed of gas diffusion layer (GDL) including gas diffusion backing (GDB) and microporous layer (MPL), membrane electrode assembly (MEA), and bipolar plates with gas channels. The fibrous gas diffusion layer is a core component of a PEMFC, enabling transport of gases, liquids and electricity within the cell. In this paper, computational and experimental advanced tools were combined to preliminary describe the relationship between the microstructure of highly sulfonated sPSU (DS = 80%) and its physico-chemical, mechanical and electrochemical features. Computer simulations allowed to describe the architecture and to estimate the structural parameters of the sPSU membrane. Molecular dynamics revealed an interconnected lamellar-like structure for hydrated sPSU, with ionic clusters of about 14-18 Å in diameter corresponding to the hydrophilic sulfonic-acid-containing phase. I am pleased to send you moderate comments. The results and theme of this paper is quite interesting. The layout is clear and easy to understand. Generally, this manuscript makes fair impression and my recommendation is that it merits publication in this Journal, after the following major revision:

  1. The authors need to reorganize the current introduction, which normally consists of three parts at least: background, literature review, brief of the proposed work. The current one is nothing but a literature review. Why their work is important comparing to previous reports? I think this is essential to keep the interest of the reader.
  2. Nonetheless, the single H2/O2 fuel cell assembled with sPSU exhibited better features than any earlier published hydrocarbon ionomers, thus opening interesting perspectives toward the design and preparation of high-performing sPSU-based PEMs. The authors should give some explanation on above results.
  3. Materials and Methods part. Although the results look “making sense”, the current form reads like a simple lab report. The authors should dig deeper in the results by presenting some in-depth discussion.
  4. In Fig.5, 7, 9 and 10, the authors should give the explanations for the difference of data collected from different sources.
  5. Proton exchange membrane fuel cells have attracted attention from energy devices such as portable, mobile and stationary devices, since it helps effective reductions of energy shortage and environment pollution. In the theoretic perspective, fractal theory is a very important tool, which can be used to investigate the physico-chemical, mechanical and electrochemical features of proton exchange membrane fuel cells, see [International Journal of Hydrogen Energy, 2018, 43(37):17880-17888; Fractals, 2019, 27(2):1950012]. Authors should introduce some related knowledge to readers.
  6. I am quite interested in some parametric study with the proposed molecular dynamics. The manuscript could be more substantial if the authors do so. At least, the authors need to write some statements that how the proposed molecular dynamics can be used for the parametric study.

Author Response

Response to Reviewer 1 Comments.

Hydrogen as an energy resource can make life significantly more eco-friendly. A hydrogen fuel cell works based on electrochemistry, by passing hydrogen through the anode and oxygen through the cathode of the cell. Proton exchange membrane fuel cell (PEMFC) which includes a membrane, and two electrodes, has grown up with huge attraction because of its simple operation and fuel availability. The proton exchange membrane fuel cells (PEMFCs) are promising energy devices for stationary and mobile applications because of high power density, high efficiency, low operating temperature, low emissions, low noise, and great environmental compatibility. The PEMFCs are composed of gas diffusion layer (GDL) including gas diffusion backing (GDB) and microporous layer (MPL), membrane electrode assembly (MEA), and bipolar plates with gas channels. The fibrous gas diffusion layer is a core component of a PEMFC, enabling transport of gases, liquids and electricity within the cell. In this paper, computational and experimental advanced tools were combined to preliminary describe the relationship between the microstructure of highly sulfonated sPSU (DS = 80%) and its physico-chemical, mechanical and electrochemical features. Computer simulations allowed to describe the architecture and to estimate the structural parameters of the sPSU membrane. Molecular dynamics revealed an interconnected lamellar-like structure for hydrated sPSU, with ionic clusters of about 14-18 Å in diameter corresponding to the hydrophilic sulfonic-acid-containing phase. I am pleased to send you moderate comments. The results and theme of this paper is quite interesting. The layout is clear and easy to understand. Generally, this manuscript makes fair impression and my recommendation is that it merits publication in this Journal, after the following major revision:

  1. The authors need to reorganize the current introduction, which normally consists of three parts at least: background, literature review, brief of the proposed work. The current one is nothing but a literature review. Why their work is important comparing to previous reports? I think this is essential to keep the interest of the reader.

Authors reply: We would like to thank the Reviewer for his/her comments on our work. Regarding the revisions, the introduction has been largely revised, trying to highlight the advances, excellence and originality of our work in this field.

  1. Nonetheless, the single H2/O2 fuel cell assembled with sPSU exhibited better features than any earlier published hydrocarbon ionomers, thus opening interesting perspectives toward the design and preparation of high-performing sPSU-based PEMs. The authors should give some explanation on above results.

Authors reply: We thank the reviewer for the clarification required. Clearly, we need to underline that the comparison with other non-fluorinated membranes is very difficult since the operating conditions used in the fuel cell tests, as well as the Pt loading at the electrodes, are often quite different. In this case, the higher cell performance of the sPSU membrane compared to analogs PEMs might be related to the very high Ion Exchange Capacity investigated during this study.

  1. Materials and Methods part. Although the results look “making sense”, the current form reads like a simple lab report. The authors should dig deeper in the results by presenting some in-depth discussion.

Authors reply: We would like to thank the reviewer for the kind suggestion. The “Materials and Methods” and the “Results & Discussion” sections have been improved and implemented with new details. We hope he/she will be satisfied with our answers and believe that the paper has become stronger as a consequence of the revision.

  1. In Fig.5, 7, 9 and 10, the authors should give the explanations for the difference of data collected from different sources.

Authors reply: As envisaged by the reviewer, the data reported in Fig. 5, 7, 9 and 10, arises from different experimental conditions. For the NMR investigation (Figure 5 and 7) the sPSU membrane was first equilibrated at four initial water content, i.e. saturation (38 wt%, maximum membrane swelling), 30 wt%, 20 wt% and 10 wt%, respectively. Thereafter, for each water content, the temperature evolution of the self-diffusion coefficient (Figure 5) and consequently of the hydration number (Figure 7) were investigated. This allowed to better elucidate the relationship between water content and water molecular dynamics in the sPSU membrane.

Contrariwise, to assess the dimensional swelling of the sPSU membrane (Figure 9), the sample was immersed in distilled water for 24 h at 20 °C and for 2 h at higher temperatures (from 40 to 100 °C each 20 °C). This is a typical procedure to test the dimensional stability of a PEM.

Finally, in the case of the EIS investigation (Figure 10), the proton conductivity was investigated at four different relative humidity (90, 70, 50 and 30 RH%) in the temperature range 20-120 °C. In this case, however, the humidity is kept constant during heating, that is also, a quite typical procedure in the field in order to obtain crucial information about the conductivity performance of the PEM under a wide range of operating conditions.

To summarize, each one of the dataset illustrated in the aforementioned Figures provides unique information on the physic-chemical properties of sPSU. This aspect is now well discussed in the revised manuscript.

  1. Proton exchange membrane fuel cells have attracted attention from energy devices such as portable, mobile and stationary devices, since it helps effective reductions of energy shortage and environment pollution. In the theoretic perspective, fractal theory is a very important tool, which can be used to investigate the physico-chemical, mechanical and electrochemical features of proton exchange membrane fuel cells, see [International Journal of Hydrogen Energy, 2018, 43(37):17880-17888; Fractals, 2019, 27(2):1950012]. Authors should introduce some related knowledge to readers.

Authors reply: We thank the reviewer for the opportunity to improve the manuscript. In the

actual form, indeed, the paper, in the Results and Discussion section, now briefly introduces

the different levels of theory that can be adopted in the study of physico-chemical properties

of polymers, in general, and in the development of fuel cells, like all-atoms and coarse

grained molecular dynamics simulations or fractal theory, as follows:

“Due to the lack of information about the internal rearrangement characterizing the structure of the sPSU-membrane, we decided to carry out all-atoms classical MD simulations in order

to investigate conformational behavior of the membrane, with the aim to deeply understand

how this can affect the efficiency in PEM-FCs. It is worth noting, indeed, that the theoretical

modelling of polymers, nowadays, plays a crucial role in the understanding of chemico-

physical properties of polymers. The information obtained from simulations can further

adopted in the design of new materials and the discovery of new synthetic routes. At this

purpose, it has been demonstrated as many levels of theory can provide useful insights in

polymers science and in the further development of FCs, in particular, adopting modern

computational chemistry tools, like Coarse Grained (CG) simulations and fractal theory (see

refs [34–36] as recent examples).”

The bibliography, consequently, has been enriched with further references, including those

suggested by the reviewer.

  1. I am quite interested in some parametric study with the proposed molecular dynamics. The manuscript could be more substantial if the authors do so. At least, the authors need to write some statements that how the proposed molecular dynamics can be used for the parametric study.

Authors reply: We thank the reviewer for the opportunity to elucidate this aspect. An

example of application for parametric study, from which MD simulation can be useful, is

reported in the manuscript. The obstruction factor has been calculated according to the

pore’s size estimated along the three molecular dynamics; this parameter, subsequently, has

been adopted to obtain bound and free water fractions (for details, see the discussion in

Section 3.2). Furthermore, we added a sentence that briefly describes the possible uses of

results obtained from MDs, as follows:

“The results obtained from the analysis of structural and dynamical parameters, such as

radial distribution function and hydration number, have been adopted in the experimental

investigation that follows in the next paragraph. Additionally, the results here highlighted

can be helpful for the set-up of more accurate force fields, for both classical and CG-MDs,

for studying of sPSU-based membranes with models with higher number of atoms, usually

more accurate in the obtainment of relevant thermodynamic parameters.”

Reviewer 2 Report

In the present work, authors reported that the exploring the structure-performance relationship of sulfonated polysulfone proton exchange membrane by a combined computational and experimental approach. Authors done both computational and experimental work. However, the experimental section is some weak. Some additional characterizations are necessary. Hence, this manuscript requires minor revision as follows.

  1. Authors should investigate the morphology of membranes by either SEM or TEM analysis.
  2. Figure 2: If possible, authors may provide the high-resolution image.
  3. Table 2: I found that authors mentioned the power density value of membranes in Table 2. However, it would be better if authors can provide the power density curve as the separate figure in main manuscript instead of supporting information.
  4. Some of the following related references should be cited: 1016/j.compositesb.2018.08.016, 10.1016/j.compositesb.2020.107890, 10.1016/j.jechem.2018.02.020.

Author Response

Response to Reviewer 2 Comments.

In the present work, authors reported that the exploring the structure-performance relationship of sulfonated polysulfone proton exchange membrane by a combined computational and experimental approach. Authors done both computational and experimental work. However, the experimental section is some weak. Some additional characterizations are necessary. Hence, this manuscript requires minor revision as follows.

  1. Authors should investigate the morphology of membranes by either SEM or TEM analysis.

Authors reply: We thank the reviewer for his/her kind suggestion. Scanning electron microscopy (SEM) was used to investigate the internal microstructure of the SPSU membrane and the results discussed in the revised version of the manuscript. In particular, we added the following:

“Figure S1 illustrates the surface and cross-sectional SEM images of the sPSU membrane. The film exhibits a dense, compact and homogeneous structure. Furthermore, the sPSU membrane presents a smooth and uniform morphology without any cracks and/or defects. This clearly proves the good quality of the prepared membranes.”

  1. Figure 2: If possible, authors may provide the high-resolution image.

Authors reply: The image has been improved.

  1. Table 2: I found that authors mentioned the power density value of membranes in Table 2. However, it would be better if authors can provide the power density curve as the separate figure in main manuscript instead of supporting information.

Authors reply: In agreement with the reviewer's observation, the polarization and power density curves for the sPSU membrane and the Nafion 212 benchmark have been moved in the main manuscript.

  1. Some of the following related references should be cited: 1016/j.compositesb.2018.08.016, 10.1016/j.compositesb.2020.107890, 10.1016/j.jechem.2018.02.020.

Authors reply: We would like to thank the reviewer for his/her comments on our work. As suggested by the reviewer, we have enriched the bibliography with other references.

Round 2

Reviewer 1 Report

In Ref. 37, “2019, 27” should be corrected as “2019, 27, 1950012”